# Cost-effectiveness of a high-intensity versus a low-intensity smoking cessation intervention in a dental setting: long-term follow-up

Inna Feldman,[1] Asgeir Runar Helgason,[2,3] Pia Johansson,[4] Åke Tegelberg,[5,6] Eva Nohlert[7]

For numbered affiliations see end of article.

**Correspondence to**
Dr Inna Feldman;
inna.feldman@pubcare.uu.se

## ABSTRACT

**Objectives** The aim of this study was to conduct a cost-effectiveness analysis (CEA) of a high-intensity and a low-intensity smoking cessation treatment programme (HIT and LIT) using long-term follow-up effectiveness data and to validate the cost-effectiveness results based on short-term follow-up.

**Design and outcome measures** Intervention effectiveness was estimated in a randomised controlled trial as numbers of abstinent participants after 1 and 5–8 years of follow-up. The economic evaluation was performed from a societal perspective using a Markov model by estimating future disease-related costs (in Euro (€) 2018) and health effects (in quality-adjusted life-years (QALYs)). Programmes were explicitly compared in an incremental analysis, and the results were presented as an incremental cost-effectiveness ratio.

**Setting** The study was conducted in dental clinics in Sweden.

**Participants** 294 smokers aged 19–71 years were included in the study.

**Interventions** Behaviour therapy, coaching and pharmacological advice (HIT) was compared with one counselling session introducing a conventional self-help programme (LIT).

**Results** The more costly HIT led to higher number of 6-month continuous abstinent participants after 1 year and higher number of sustained abstinent participants after 5–8 years, which translates into larger societal costs avoided and health gains than LIT. The incremental cost/QALY of HIT compared with LIT amounted to €918 and €3786 using short-term and long-term effectiveness, respectively, which is considered very cost-effective in Sweden.

**Conclusion** CEA favours the more costly HIT if decision makers are willing to spend at least €4000/QALY for tobacco cessation treatment.

## INTRODUCTION

Smoking is likely to remain the single most important preventable health risk in the world. Despite continuously declining prevalence in recent decades, 1 in 10 adults in Sweden still smokes daily.[1] Cigarette smoking contributes to 7.5% of the burden of disease

### Strengths and limitations of this study

► This study uses a unique possibility to compare cost-effectiveness analyses based on 1 and 5–8 years of follow-up data.
► This economic evaluation clearly supports that more intensive and costly smoking cessation provision is cost-effective.
► The calculation of the intervention costs for the cessation programmes was based on a trial protocol and might be overestimated in comparison with routine practice.
► The effects of smoking cessation are probably underestimated since only three disease groups are modelled and no effects of passive smoking are included.

in Sweden[2] and was estimated to account for approximately €3 000 000 (31.5 billion Swedish kronor, SEK), including €1 000 000 (11 billion SEK) in healthcare costs (15% of the national costs for health and welfare sector) and €1 500 000 (16 billion SEK) in productivity costs in year 2015.[3] A decrease in prevalence of smoking to 5% could save society €1 300 000 (14.3 billion SEK) per year.

Several smoking cessation interventions, targeted at current smokers, are available; furthermore, evaluations so far have confirmed the effectiveness of the majority of them. Additionally, some recent studies emphasise that higher level of intervention intensity, such as additional counselling sessions[4] and intensive support through a mobile application,[5] resulted in the highest smoking cessation rates. However, due to increasing number of available interventions, decision makers have to decide which intervention to implement, taking into account that intervention intensity increases intervention costs. Relative costs and benefits of those interventions are important criteria,

thus, increasing the attention on economic evaluations in recent years.[6 7] Economic evaluations combine the costs and outcomes of different interventions and aim to determine which intervention provides the best value for money.[8] Several studies on the cost-effectiveness of smoking cessation interventions comparing different intensity of support have been performed during the last few years, for example, Quit-and-Win programme,[9] comparison of standard, enhanced and intensive smoking cessation interventions using cell phones,[10] and two smoking cessation approaches of different level of intensity for patients with cancer.[11] The results suggested that the higher intensive interventions are preferable from health economics point of view, but all those evaluations were based on 6-month or 12-month follow-up; long-term follow-ups are scarce in randomised controlled trials (RCTs).

The effects of smoking on health occur during many years because current smoking influences future health risks; similarly, a smoking cessation today will cause smoking-related health risks to tail off gradually. Thus, in order to estimate cost-effectiveness of smoking cessation interventions, a lifetime perspective is necessary, taking into account a variety of different costs and effects.[12] Hence, the well-established method to perform cost-effectiveness analyses (CEAs) of smoking cessation interventions involves mathematical modelling of future events as consequences of smoking. Systematic reviews of model-based economic evaluations in smoking cessation analysed different aspects, such as type of model, quality of the model, transferability and comparison of the results in different studies.[12–14] Berg *et al*[13] identified 64 economic evaluations in smoking cessation, and the state-transition Markov model was the most frequently used. The majority of the models simulates the lifetime development of morbidity and mortality for smoker versus former smoker using relative risks for four diseases: chronic obstructive pulmonary disease (COPD), coronary heart disease (CHD), stroke and lung cancer. The authors concluded that existing economic evaluations in smoking cessation vary in quality, resulting mainly from the way in which they selected their populations, measured costs and effects, and assessed the variability and generalisability of their own findings.[13] One of the reasons of the quality issues is that all those studies are based on short-term follow-up (from 6 months to 1 year), and they have no possibilities to validate the sustainability of short-term effectiveness in real life; thus, they cannot confirm the reported cost-effectiveness results and policy recommendations. Moreover, the long-term assumption, such as relapse rate, might change the results of the smoking cessation cost-effectiveness.[15]

Our previous economic evaluation of high-intensity and low-intensity treatment programmes (HIT and LIT) for smoking cessation in a dental setting was based on the reported number of quitters measured as point prevalence abstinent (not one puff of smoke during the past 7 days prior to 1-year follow-up). The conclusion was that high-intensity treatment support is the preferred option if the decision makers' willingness-to-pay exceeds €5100 (50 000 SEK) per QALY. The base-case scenario of the analysis assumed a sustained abstinence for the quitters.[16] The long-term follow-up of the programmes was performed 5–8 years later.[17] In this study, we used a unique opportunity to compare CEAs of a high-intensity and a low-intensity smoking cessation intervention in a dental setting using data from short-term (1 year) and long-term (5–8 years) follow-up.

We set out to: (1) perform a cost-effectiveness analysis of a high-and a low-intensity smoking cessation programme in a dental setting using long-term (5–8 years) follow-up data and (2) compare the cost-effectiveness results with the previous study based on short-term (1-year) follow-up.

## METHODS
### Summary of the smoking cessation study
In the smoking cessation intervention study,[18] between August 2003 and February 2005, 300 adult smokers recruited via direct inquiry or advertising in dental or general healthcare were offered smoking cessation support performed in a dental setting. Inclusion criteria were daily smokers over 20 years of age, while exclusion criteria were reading difficulties and problems with Swedish language. The participants were randomly assigned to two interventions; one received high-intensity treatment support and one low-intensity treatment support.

The high-intensity smoking cessation treatment, the HIT programme, comprised eight individual sessions, 3.5 hours in total over a period of 4 months, and was based on behaviour therapy, coaching and pharmacological advice. The low-intensity smoking cessation treatment, the LIT programme, comprised one counselling session, of up to 45 min, introducing a conventional self-help programme running over 8 weeks. Both programmes were free of charge.

The participants answered a baseline questionnaire and a short-term (1 year after the planned smoking cessation date) follow-up questionnaire. Demographic characteristics such as gender, age and education level were also collected. The effectiveness of the trial was reported elsewhere.[18] The analysis concluded that the more extensive and expensive HIT programme was more effective and cost-effective in terms of proportion of smokers who were still smoke-free after 1 year.[16 18] The long-term follow-up was performed 5–8 years after the planned smoking cessation date. The effectiveness analysis showed that the difference in outcome between the HIT and LIT programmes remained relatively constant and significant in favour of HIT, and that abstinence at 1-year follow-up was a good predictor for long-term abstinence.[17] All analyses were done using the 'intention to treat' approach where non-responders were considered as smokers. Mortality and morbidity data for the participants were not collected either by questionnaire or through the registers.

The mean age of the participants was 49 years, and 78% were women. Short-term follow-up (1 year) questionnaire was answered by 84% of the randomised participants (88% for HIT vs 81% for LIT). Fourteen per cent (41 of the 300 participants) reported 6-month continuous abstinence (not one puff of smoke during the past 6 months); 27 (18%) individuals in HIT versus 14 (9%) in LIT. At long-term follow-up (5–8 years), 241 persons answered the questionnaire (80% for both HIT and LIT). Of those, 24 were sustained abstinent (17 vs 7 for HIT vs LIT) since the planned smoking cessation date. Relapse rate was 26% and 50% for participants reported 6-month continuous abstinence at 1-year follow-up in HIT and LIT, respectively, but the difference was not statistically significant. Characteristics of the study participants, as well as abstinence at the 1 year and at the long-term follow-up, are presented in table 1.

## Economic evaluation

Two economic evaluations were performed to obtain the cost-effectiveness of the more costly HIT programme in comparison to LIT:

1. Cost-effectiveness analysis based on the number and characteristics of 6-month continuous abstinent participants according to 1-year follow-up, CEA short-term.
2. Cost-effectiveness analysis based on the number and characteristics of sustained abstinent participants since planned smoking cessation date according to 5–8 years of follow-up, CEA long-term.

Both analyses used the same methodology described below.

Economic evaluations were based on the costs to implement the programmes, the number and characteristics of abstinent participants and on a previously constructed Markov model that estimates the future health and cost consequences of smoking cessation. All costs were inflated to reflect 2018 costs according to the Swedish consumer price index[19] and converted into 2018 Euro (€) using the purchasing power parity estimates with CCEMG–EPPI-Centre Cost Converter (http://eppi.ioe.ac.uk/costconversion/default.aspx). The CEAs followed Swedish and international recommendations: costs were calculated from a societal perspective, health effects expressed as quality-adjusted life-years (QALYs) and programmes explicitly compared in an incremental analysis (incremental cost-effectiveness ratio (ICER)), with discounting (3% per year) and sensitivity analyses.[8 20] The ICER was calculated by dividing the difference in total costs for the programmes (incremental cost) by the difference in the health outcomes in QALYs (incremental effect) to provide a ratio of extra cost per extra unit of health effect.

## Intervention costs

The intervention costs were collected prospectively by interviewing the three dental hygienists who carried out the patient work, as well as the project leader and the project coordinator. The costs were divided into joint costs for the two programmes and programme-specific costs,

**Table 1** Characteristics of the study participants and programme effectiveness at 1 and 5–8 years of follow-up, by treatment intensity

| | HIT N=150 | LIT N=150 | P value |
|---|---|---|---|
| Study participants (n) | | | |
| Baseline measures | 146 | 148 | |
| 12-month follow-up measures | 132 | 122 | |
| Available at long-term follow-up | 141 | 143 | |
| Long-term follow-up measures | 121 | 120 | |
| Participants characteristics | | | |
| Gender (n) | | | |
| Men | 26 | 32 | 0.410 |
| Women | 115 | 111 | |
| Age at baseline (age) | | | |
| Mean (SD) | 48.7 (9.6) | 48.5 (11.0) | 0.825 |
| Median | 48.0 | 49.0 | |
| Education (in years) (n) | | | |
| 0–9 | 25 | 36 | 0.336 |
| 10–12 | 60 | 55 | |
| >=13 | 52 | 50 | |
| Smoked cigarettes/ week at baseline (n) | | | |
| Mean (SD) | 106 (50) | 105 (40) | 0.794 |
| Median | 105 | 105 | |
| Intervention effectiveness (n) | | | |
| 1-year follow-up | | | |
| 6-month continuous abstinence | 27 | 14 | 0.034* |
| 5–8 years of follow-up | | | |
| Sustained abstinence | 17 | 7 | 0.030* |
| Relapse rate (%) | 26 | 50 | 0.345 |

*Statistical significant differences at 0.05 level in effectiveness between the programmes.
HIT, High-intensity smoking cessation treatment; LIT, Low-intensity smoking cessation treatment.

and undiscounted because of the short 3-year project time. The joint costs were assumed, divided equally between the programmes, while the programme-specific costs included staff time for patient work, material and participant costs. Estimation of the intervention costs has been described in detail previously.[16] Total programme-specific costs amounted to €117011; €801 per participant for HIT and €27927; €189 per participant for LIT.

**Table 2** Logistic regression analysis of factors associated with sustained abstinence at 5–8 years of follow-up

|  | Coefficient | P value | OR | 95% CI |
| --- | --- | --- | --- | --- |
| HIT programme | 1.001 | 0.03* | 2.72 | 1.09 to 6.80 |
| Male gender | −0.077 | 0.88 | 0.93 | 0.32 to 2.64 |
| Age | 0.005 | 0.82 | 1.00 | 0.96 to 1.05 |
| Constant | −3124 | 0.01 | 0.04 |  |

*Statistically significant at 0.05 level.
HIT, High-intensity smoking cessation treatment; OR, Odds Ratio.

### Intervention effectiveness

For CEA short term, we used 6-month continuous abstinence at 1-year follow-up reported by 41 participants (14 from HIT and 27 from LIT). For CEA long term, we used sustained abstinence at 5–8 years reported by 24 participants (17 from HIT and 7 from LIT), see table 1. Both measures were statistically significant different between the treatment programmes. In order to generalise the long-term effectiveness of our study, we performed a logistic regression analysis to calculate the probability of sustained abstinence depending on programme (HIT vs LIT), participant's gender and age, see table 2.

The type of the programme (HIT vs LIT) was significantly associated with sustained abstinence, while gender and age were not. The regression equation (1) demonstrates dependence between 'abstinence' (1—abstinence, 0—no abstinence) and 'programme' (1—HIT, 0—LIT), 'gender' (1—male, 0—female) and 'age' (19-71):

$$abstinence = -3.124 + 1.001 * programme - 0.077 * gender + 0.005 * age \quad (1)$$

Equation (1) allows us to calculate the probability of long-term abstinence, $P_q$, for a random participant (a random man/woman from a population of interest, smoker between 19 and 71 years old) in respective programme, see equation (2),

$$P_q = EXP\ (abstinence)/(1 + EXP\ (abstinence))\ (2)$$

### Markov model

A Markov model was used to estimate health consequences and societal costs of smoking cessation, further described in a technical report.[21] The model has been used in similar studies in Sweden,[16 22 23] and the updated year 2015 version was used for the current analysis.[21] The model simulates the societal effects of quitting smoking on three disease groups: lung cancer, COPD and cardiovascular disease, including CHD and stroke. Even though there are other smoking-related diseases, these conditions cover most of the health problems associated with smoking.[24] The model incorporates the smoking-related disease risks, time-dependent remaining excess disease risks after quitting, the death risks for the specific and for unrelated diseases, as well as the societal costs of the diseases. All disease risks are annual age-specific and gender-specific excess incidence risks until death or the age of 95. This lifetime horizon was recommended for modelling of smoking cessation interventions[12] because smoking cessation reduces smoking-related health risks gradually during a long period. Notably, the model does not contain the risk for relapse in smoking among the quitters. The societal costs include costs associated with medical treatment, community care, drugs, informal care and other expenditures for patients and relatives, as well as morbidity productivity costs. Health outcomes are expressed in QALYs. The number of QALYs was calculated during healthy years and years spent with a disease, until death or the age of 95. The model and all the parameters are described in detail in a technical report[21] and online supplementary appendix 1.

Model simulations were performed according to gender and 5-year age groups. The simulations result in accumulated societal costs and health effects for lifelong continuing smokers and quitters at a specific age and gender group, respectively. The differences in societal costs and health effects between smoking statuses at a certain age are then compared outside the model, and constitute the avoided costs and gained health effects from the tobacco quitting for the specified age and gender group.

### Sensitivity analyses

Extensive sensitivity analyses on parameter values and methodological choices were reported in the model technical report.[21] The model estimates were, in general, insensitive to changes in parameter values, except the most conservative multivariate analysis where the costs were decreased by 25%, the disease risks by 50%, the death risks by 10% and the risk fractions after quitting by 0.1. This low cost/low risk analysis led to substantial decreases in cost and QALY differences between quitters and smokers. This sensitivity analysis was applied to compare costs and effects between HIT and LIT, to validate the results of the CEA long-term.

To increase the generalisability of the cost-effectiveness results, we have also applied the probabilities of long-term abstinence depending on programme (HIT vs LIT), participant's gender and age on the modelling results. We estimated the avoided social costs and gained QALYs for a random quitter from our sample and then adjusted the results to the probability to quit (abstinence), calculated in (1). Cost-effectiveness was estimated for men and women separately.

Further, a probabilistic sensitivity analysis (PSA) was conducted, based on the uncertainty of the difference in sustained abstinent participants in the two programmes. The effectiveness of LIT was fixed at the 7% quit rate, but the HIT quit rate was sampled from the 95% CI (9%–22%). The PSA was performed by 10 000 runs using the societal costs avoided and QALYs gained for the group with the largest number of quitters, that is, women aged 40–44 years. The PSA was presented as a cost-effectiveness

**Table 3** Model estimates of societal costs avoided and QALYs gained. Costs in Euro 2018. 3% discount rate

| Gender/Age group | Model estimates Costs avoided | QALYs§ gained | CEA short* HIT† NHp¶ | LIT‡ NLp** | Difference N†† | Costs | QALYs§ | CEA long* HIT† NHs‡‡ | LIT‡ NLs§§ | Difference N¶¶ | Costs | QALYs§ |
|---|---|---|---|---|---|---|---|---|---|---|---|---|
| Women | | | | | | | | | | | | |
| 20–24 | 8142 | 0.61 | | 1 | −1 | −8142 | −0.61 | na | na | na | na | na |
| 25–29 | 8425 | 0.65 | 1 | | 1 | 8425 | 0.65 | na | na | na | na | na |
| 35–39 | 9267 | 0.71 | 2 | 2 | 0 | | | 1 | 1 | 0 | | |
| 40–44 | 8532 | 0.71 | 5 | | 5 | 42658 | 3.55 | 4 | | 4 | 34126 | 2.84 |
| 45–49 | 6772 | 0.66 | 3 | 3 | 0 | | | 1 | 2 | −1 | −6772 | −0.66 |
| 50–54 | 5228 | 0.61 | 4 | 3 | 1 | 5228 | 0.61 | 1 | 2 | −1 | −5228 | −0.61 |
| 55–59 | 4542 | 0.43 | 4 | 2 | 2 | 9085 | 0.86 | 4 | 1 | 3 | 13627 | 1.29 |
| 60–64 | 3336 | 0.32 | 4 | | 4 | 13342 | 1.29 | 2 | | 2 | 6671 | 0.64 |
| 65–69 | 2023 | 0.33 | | 1 | −1 | −2023 | −0.33 | na | na | na | na | na |
| Men | | | | | | | | | | | | |
| 20–24 | 10430 | 0.74 | 1 | | 1 | 10430 | 0.74 | 1 | | 1 | 10430 | 0.74 |
| 40–44 | 10526 | 1.00 | 1 | | 1 | 10526 | 1.00 | 1 | | 1 | 10526 | 1.00 |
| 45–49 | 11416 | 0.82 | 1 | 1 | 0 | | | 1 | 1 | 0 | | |
| 50–54 | 11360 | 0.78 | | 1 | −1 | −11360 | −0.78 | na | na | na | na | na |
| 65–69 | 4084 | 0.46 | 1 | | 1 | 4084 | 0.46 | 1 | | 1 | 4084 | 0.46 |
| Total | | | 27 | 14 | 13 | 82253 | 7.44 | 17 | 7 | 10 | 67466 | 5.71 |

*Cost-effectiveness analysis.
†High-intensity smoking cessation treatment, the HIT programme.
‡Low-intensity smoking cessation treatment, the LIT programme.
§Quality-adjusted life-years.
¶NHp—number of 6-month continuous abstinent participants HIT treatment programme according to 1-year follow-up.
**NLp—number of 6-month continuous abstinent participants LIT treatment programme according to 1-year follow-up.
††Difference in numbers of 6 -month continuous abstinent participants between the treatment programmes according to 1 -year follow-up.
‡‡NHs—number of sustained abstinent participants HIT treatment according to 5–8 years of follow-up.
§§NLs number of sustained abstinent participants LIT treatment according to 5–8 years of follow-up.
¶¶Difference in number of sustained abstinent participants between the treatment programmes according to 5–8 years of follow-up.
na, not applicable.

acceptability curve, which indicates the probability that HIT is cost-effective versus LIT at different values of the willingness-to-pay for a QALY.

### Patient and public involvement

This research was done without patient involvement. Patients were not invited to comment on the study design and were not consulted to develop patient relevant outcomes or interpret the results. Patients were not invited to contribute to the writing or editing of this document for readability or accuracy.

### RESULTS
### Model estimates

Model estimates for the CEA short term and CEA long term are presented in table 3 (societal costs and QALYs). The second and third columns in table 3 present the estimation of avoided societal costs and QALYs gained for a person with respective gender and age, who became

sustained abstinent in comparison with a continuing smoker. Using these data, we can estimate the difference in societal cost avoided and QALYs gained by multiplying difference in numbers of 6-month continuous abstinent participants between the treatment programmes (N*) or difference in numbers of sustained abstinent participants since planned smoking cessation date between the treatment programmes (N**) by societal costs avoided and QALYs gained.

The CEA short-term indicated that HIT led to additional avoided societal costs of €82253 and additional 7.44 QALYs compared with LIT. The CEA long- term reported the difference between HIT and LIT as additional avoided societal costs of €67466 and additional 5.71 QALYs.

### Cost-effectiveness analyses

The more costly HIT programme led to a higher number of 6-month continuous abstinent participants at 1-year

**Table 4** Incremental cost-effectiveness analyses (CEAs) of the two smoking cessation treatments, HIT and LIT, for 6-month continuous abstinence at 1 year (CEA short term), sustained abstinence at 5–8 years of follow-up (CEA long term), and sensitivity analyses for CEA long term. Societal perspective, in Euro 2018

| Intervention costs | CEA* short | CEA* long | CEA* long, sensitivity | CEA* long, population level, per person | |
|---|---|---|---|---|---|
| | | | | Men | Women |
| HIT† | 117 011 | 117 011 | 117 011 | 801 | 801 |
| LIT‡ | 27 927 | 27 927 | 27 927 | 189 | 189 |
| Difference in intervention costs | 89 085 | 89 085 | 89 085 | 612 | 612 |
| Difference in societal costs avoided | 82 253 | 67 466 | 32 469 | 779 | 502 |
| Incremental costs | 6832 | 21 619 | 56 616 | −167 | 110 |
| Incremental QALYs§ | 7.44 | 5.71 | 4.82 | 0.0664 | 0.0462 |
| Incremental cost per QALY§ (ICER¶) | 918 | 3786 | 11 746 | <0 | 2391 |

*Cost-effectiveness analysis.
†High-intensity smoking cessation treatment, the HIT programme.
‡Low-intensity smoking cessation treatment, the LIT programme.
§Quality-adjusted life-years.
¶Incremental cost-effectiveness ratio (ICER) is calculated as incremental costs divided by incremental quality-adjusted life-years (QALYs).

follow-up (CEA short-term), as well as higher number of sustained abstinent participants at 5–8 years of follow-up (CEA long -erm), which translates into larger costs avoided and health gains than LIT, see table 4. However, the difference in intervention costs were not fully balanced by the societal costs avoided, so HIT implied an incremental net cost of about €6832 in CEA short -term and €21 619 in CEA long-term, compared with LIT. HIT was estimated to lead to more QALYs, so the incremental cost per QALY of HIT compared with LIT amounted to €918 for CEA short-term and €3786 for CEA long-term, which is considered to be very cost-effective in Sweden.[20] The incremental analysis favours the more costly HIT, if decision makers are willing to spend at least €4000/QALY for tobacco cessation programmes.

### Sensitivity analyses

The most conservative sensitivity analysis, a multivariate low cost/low risk analysis, was applied to CEA long-term. This analysis led to substantial decreases in avoided social costs and QALY gains for both HIT and LIT. At the same time, the incremental costs increased and incremental QALYs slightly decreased which resulted in higher incremental of €11 746 per QALY, see table 4.

The probability of sustained abstinence varies between 0.11 and 0.13 for men and between 0.12 and 0.14 for women in HIT in different ages. The corresponding numbers are 0.4–0.5 for men and 0.5–0.6 for women in LIT. The model estimates for random men and women were €9740/0.83 and €7165/0.66 for avoided societal costs/QALYs gained. Given the probability of abstinence, the difference in avoided societal costs per person between HIT and LIT was estimated as €779 for men and €502 for women and the correspondent difference in QALYs gained was 0.0664 for men and 0.0462 for women. The ICER was negative for men (HIT was cost saving and entailed positive health outcomes in comparison to LIT)

but amounted to €2391 for women, which is close to our base-case analysis, see table 4.

At all values of willingness-to-pay for a QALY, including zero, the HIT was more cost-effective than the LIT, see the PSA on the HIT quit rate in figure 1.

## DISCUSSION
### Main results

In this study, we performed a a cost-effectiveness analysis using the long-term follow-up data from an RCT of a high-intensity and a low-intensity treatment programme (HIT and LIT) for smoking cessation in a dental setting. We also validated the cost-effectiveness results of the previous study based on short-term follow-up.[16] HIT was more effective in getting participants to quit smoking and to keep sustained abstinent, resulted in higher societal costs avoided and more QALYs gained among both men

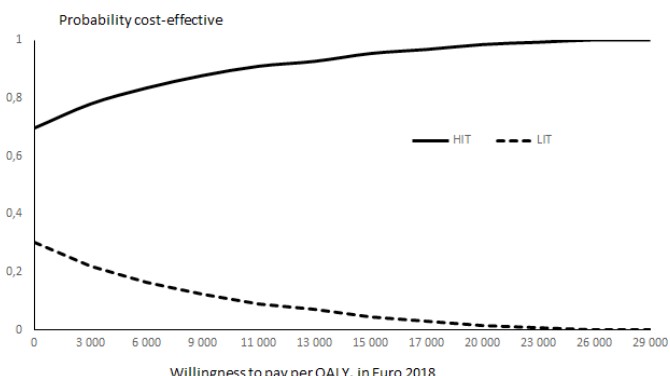

**Figure 1** Probabilistic sensitivity analysis on the effectiveness (proportion of quitters) of high-intensity treatment (HIT) in comparison with low-intensity treatment (LIT), reported as cost-effectiveness acceptability curve, willingness-to-pay per quality-adjusted life-year (QALY), in Euro 2018.

and women, compared with LIT and thus can be considered cost-effective. The ICERs were €918 and €3786 using short-term and long-term effectiveness, respectively, which are below the Swedish willingness-to-pay threshold of €50 000 per QALY,[25] thus, indicating that the resource intensive HIT was cost-effective compared with the less resource demanding LIT. The results also confirm the conclusions of the previous CEA based on short-term follow-up data and suggest its sustainability. We would recommend the use of the HIT programme as a cost-effective option for smoking cessation.

Notably, the usage of both the HIT and the LIT programmes is not limited to dental settings and can be implemented in other healthcare sectors and delivered by trained nurses instead of dental hygienists. Since the salaries of registered nurses and dental hygienists are comparable, the conclusion of high cost-effectiveness of the HIT programme remains.

However, although HIT was shown to be cost-effective in comparison with LIT, the sensitivity analysis using the probability of abstinence suggested that HIT dominated over LIT for men (saved societal costs and generated more QALYs). In our sample, the majority of study participants were women, that is why the results of the sensitivity analysis for women were very close to our base-case analysis.

### Strength and limitations

The majority of CEAs on smoking cessation use 1-year quit rates in their models; however, it is not uncommon that 6-month quit rates are used.[12 26] The question of how much we can trust the overall conclusions of such analyses always remains because we do not know for sure what happens subsequently. To our knowledge, this is the first study that uses a unique possibility to compare a previously conducted CEA based on 6-month continuous abstinent participants at 1-year follow-up with a new evaluation, based on sustained abstinence since the planned smoking cessation date up to 5–8 years. We had the possibility to compare the results based on 6-month continuous abstinence (when some time-dependent excess disease risks remained for the first years after quitting) and sustained abstinence for 5–8 years (when the smoking-related excess disease risks had been reduced). A higher proportion of sustained abstinent participants in HIT compared with LIT contributed to a low ICER for the long-term CEA.

The effects of smoking cessation are certainly underestimated in the model estimates since only three disease groups including lung cancer are modelled and no effects of passive smoking are included, but smoking is causally related to at least 15 other types of cancer. In addition, quitting smoking reduced the rate of incidence diabetes to that of non-smokers after 5 years in women and after 10 years in men.[27] The model does not include the health problems related to passive smoking, such as risk of CHDs in offspring[28] and increase in risk for breast cancer.[29] That makes our estimations more conservative with respect to

cost savings and QALYs, although these three diseases groups do account for over 80% of morbidity (and mortality) associated with smoking and are frequently used in similar studies.[15 30] Another limitation is that the model does not include the relapse rate among the quitters. This tends to overestimate the health and cost consequences of the tobacco quitting based on short-term outcomes because the relapse rate is presumably higher among the short-term quitters. On the other hand, the relapse rate might be negligibly low among individuals that quit smoking 5–8 years ago and thus not important for the modelling results. Additionally, as mentioned in our previous study,[16] the Markov model indicates considerably lower smoking-related disease risks for women reported by large epidemiological studies (see model technical report for details),[21] and thus lower cost savings and health gains from tobacco cessation for women than for men. Finally, the intervention costs for the RCT study calculation was based on the trial protocol and might be overestimated in comparison with routine practice; however, in the ICER, those extra costs were divided equally between the programmes, and thus disregarded.

### Comparison with other studies

We could not find any CEAs based on more than 1-year follow-up, and therefore we compared our results with other studies estimating cost-effectiveness of interventions with different level of intensity using 6-month or 12-month follow-up. Thus, a CEA of high-intensity multiple contest and low-intensity enhanced contest of a Quit-and-Win programme reported that high intensity Quit-and-Win leads to an average gain of 0.03 QALYs and was cost saving, in comparison with lower intensity.[9] Another study presented a CEA of three smoking cessation interventions with different intensity levels: Standard Care (SC) (brief advice to quit, nicotine replacement therapy and self-help written materials), Enhanced Care (EC) (SC plus cell phone-delivered messaging) and Intensive Care (IC) (EC plus cell phone-delivered counselling).[10] The overall conclusion was that the higher intensive intervention (IC) was the most cost-effective strategy both for men and women, which is in line with our results. Additionally, a CEA of two smoking cessation approaches for patients with cancer was presented in a study from Canada.[11] The basic programme consisted of screening for tobacco use, advice and referral, whereas the best practice programme included a basic programme and pharmacological therapy, counselling and follow-up. The ICER of the best practice programme compared with the basic programme was $3367 per QALY gained for men and $2050 per QALY gained for women. These results are very similar to our findings. In our previous study,[16] based on the same RCT and 1-year follow-up, a higher ICER of €9900/QALY and €5500/QALY was calculated for point prevalence and continuous abstinence, respectively, but the overall conclusion confirmed the cost-effectiveness of HIT at a willingness-to-pay of €10 000.

## CONCLUSIONS

In conclusion, the more costly HIT smoking cessation programme is an economically attractive option when compared with the LIT programme over a broad range of assumptions, using shot-term and long-term outcomes. CEA favours the more costly HIT if decision makers are willing to spend at least €4000/QALY for tobacco cessation treatment. These findings can support and guide implementation of smoking cessation programmes.

**Author affiliations**
[1]Department of Public Health and Caring Science, Uppsala Universitet, Uppsala, Sweden
[2]Department of Public Health Sciences, Social Medicine, Karolinska Institutet, Stockholm, Sweden
[3]Reykjavik University and Icelandic Cancer Society, Reykjavik University, Reykjavik, Iceland
[4]Public Health and Economics, Stockholm, Sweden
[5]Centre for Clinical Research, Uppsala University, Hospital of Vastmanland, Västerås, Sweden
[6]Faculty of Odontology, Malmö University, Malmö, Sweden
[7]Centre for Clinical Research, Uppsala University and Region Vastmanland, Västerås, Sweden

**Contributors** IF and EN conceived and designed the study and drafted the manuscript. Modelling and economic evaluation was carried out by IF and PJ. ARH, ÅT and EN were responsible for clinical evaluation of the smoking cessation study. All the authors (IF, ARH, ÅT, PJ and EN) contributed to the writing process and have approved the final manuscript.

**Funding** This study was funded by grants from the County Council of Västmanland, Sweden (LTV 3999) and Swedish Research Council for Health, Working Life and Welfare (FORTE), grant number 2014-1399.

**Competing interests** None declared.

**Ethics approval** The original study, as well as the long-term follow-up, was approved by the Ethical Committee at Uppsala University (Dnr:Ups 02–457, Dnr: 2010/172).

**Provenance and peer review** Not commissioned; externally peer reviewed.

**Data availability statement** Data are available on reasonable request.

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
