## [Reviewer comments · BMJ Open]

ARTICLE DETAILS

TITLE (PROVISIONAL)	Cost-effectiveness of a high- vs a low-intensity smoking cessation intervention in a dental setting: long-term follow up
AUTHORS	Feldman, Inna; Helgason, Asgeir; Johansson, Pia; Tegelberg, Åke; Nohlert, Eva

VERSION 1 – REVIEW

REVIEWER	Matthew Taylor York Health Economics Consortium, University of York, UK My employer has received funding from the National Institute for Health and care Excellence (NICE) to develop several cost-effectiveness models around smoking cessation and prevention.
REVIEW RETURNED	03-May-2019

GENERAL COMMENTS	The paper is well-written and presented very clearly for the most part. It contains a very useful analysis of the long-term follow up of a smoking cessation study. This is important because most smoking cessation models are based on data at four weeks, or six months at best. Whilst the paper is very useful, it could be improved if the following points are addressed: 1. Why are cost data for 2014 used? This should be updated to at least 2018 data, since that information would now be available.2. The authors should justify why only 1,000 iterations were used in the probabilistic sensitivity analysis. In my experience, this is rarely enough to produce 'stable' results.3. The way that the results are presented (by age and gender) is confusing. Because it is based on specific patients from the trial, many age/gender groups do not show any change at all. This could be misleading, since we know that would not be generalisable to the whole population. It would be better to perform some simple regression analysis to link age and gender with a probability of abstinence, and then model each age/gender group appropriately.4. The discussion section should provide more description of the likely biases that may have been introduced due to the various assumptions. For example, whilst the authors are correct in stating that the included health states (e.g. lung cancer, COPD, CVD) capture most of the relevant costs, they should also explain that, by leaving out other comorbidities, the model may underestimate the true benefits of smoking cessation. Similar assumptions should also be critiqued in more detail.
---

REVIEWER	Marlon Mundt University of Wisconsin School of Medicine and Public Health, USA
REVIEW RETURNED	16-May-2019

GENERAL COMMENTS	This article examines cost-effectiveness of a smoking cessation intervention offered in a dental setting. The analysis compares two treatments, one a high-intensity program of 8 sessions over 4 months and the other a low-intensity program of a single counseling session followed by self-help enrollment. The authors model cost and effectiveness based on smoking cessation rates at 5-8 years. The 5-8 year follow-up is a vast improvement over prior research and the cost-effectiveness of the intervention at an extended follow-up is of great interest. That said, there are several questions to be addressed in the study methods and results which would increase enthusiasm for the manuscript. 1. The authors have contacted the study participants at 5-8 year follow-up to assess smoking cessation. Presumably, that means that they have mortality and morbidity data for the participants. Why, then, are the mortality and morbidity findings simulated from the literature based on current smoker and prior smoker status? Why wouldn't they use the actual mortality observed in the trial? In addition, were data collected at 5-8 year follow-up on rates of COPD, lung cancer, and cardiovascular disease? If so, why were these data not used in the cost-effectiveness evaluation? If not, this seems like a missed opportunity. 2. It is not clear what assumptions were made in the cost-effectiveness modeling regarding smoking relapse. The smoking cessation rate was 18% for HIT at 1-year follow-up and 14% at 5-8 year follow-up. What assumptions were made about the timing of relapse? Did the model only consider the 14% rate? Did the model assume all 1-year abstainers immediately relapsed? Were data collected from the participants on the timing of relapse? 3. It is not clear why the Markov model is until age 95 or death. A 5-8 year time frame for the cost-effectiveness would be a welcome addition to the literature. 4. The distinction between the short-term CEA and long-term CEA is not well described. Is the only difference between the two models the quit rates? Are there assumptions in the short-term CEA about relapse rates?
--

VERSION 1 – AUTHOR RESPONSE

Answers to reviewers.

Rev. 1

1. Why are cost data for 2014 used? This should be updated to at least 2018 data, since that information would now be available.

The cost data is now updated to 2018 level. That is why the results also have been changed.

2. The authors should justify why only 1,000 iterations were used in the probabilistic sensitivity analysis. In my experience, this is rarely enough to produce 'stable' results.

We have rerun the PSA with 10, 00 iterations and changed the figure 1.

2. The way that the results are presented (by age and gender) is confusing. Because it is based on specific patients from the trial, many age/gender groups do not show any change at all. This could be misleading, since we know that would not be generalisable to the whole population. It would be better to perform some simple regression analysis to link age and gender with a probability of abstinence, and then model each age/gender group appropriately.

We have restructured the presentation of the results (Table 3) to show the only changes between the HIT and LIT programmes. We agree with the reviewer that our results are not generalizable for the whole population. The main purpose of the study was to compare HIT and LIT programmes based on short- and long-term follow up and to check if cost-effectiveness analysis based on short-term outcomes is sustainable in long-term. However, we used the reviewer's suggestion as a part of sensitivity analyses and calculated cost-effectiveness on population level, separately for men and women, see page 11, line 5-16 in Method section, page 17, line 21-23, page 18, line 1-7 in Results section and p 19, line 9-13 in Discussion section.

3. The discussion section should provide more description of the likely biases that may have been introduced due to the various assumptions. For example, whilst the authors are correct in stating that the included health states (e.g. lung cancer, COPD, CVD) capture most of the relevant costs, they should also explain that, by leaving out other comorbidities, the model may underestimate the true benefits of smoking cessation. Similar assumptions should also be critiqued in more detail.

This part was developed in the "Discussion" – section, see p. 20, line 6-13

Rev 2.

1. The authors have contacted the study participants at 5-8 year follow-up to assess smoking cessation. Presumably, that means that they have mortality and morbidity data for the participants. Why, then, are the mortality and morbidity findings simulated from the literature based on current smoker and prior smoker status? Why wouldn't they use the actual mortality observed in the trial? In addition, were data collected at 5-8 year follow-up on rates of COPD, lung cancer, and cardiovascular disease? If so, why were these data not used in the cost-effectiveness evaluation? If not, this seems like a missed opportunity.

Unfortunately, mortality and morbidity data for the participants were not collected either by questionnaire or through the registers because this study was not planned for economic evaluation, which is why we use modelling as the only possible way to estimate cost-effectiveness of the programmes. We add a sentence to clarify that (p. 7, line 10-11, 19-20)

2. It is not clear what assumptions were made in the cost-effectiveness modeling regarding smoking relapse. The smoking cessation rate was 18% for HIT at 1-year follow-up and 14% at 5-8 year follow-up. What assumptions were made about the timing of relapse? Did the model only consider the 14% rate? Did the model assume all 1-year abstainers immediately relapsed? Were data collected from the participants on the timing of relapse?

The model does not contain the risks for relapse to smoking among the quitters. This tends to overestimate the health and cost consequences of the tobacco quitting, dependent on the extent of relapse among the quitters. The relapse rate is presumably higher among the short-term quitters than among the long-term quitters, and is probably very low among individuals that quit smoking 5-8 years ago. Relapse rate were calculated for both programmes and the difference was not statistically significant (see p.8, line 4-6). Because the main purpose of the analyses was to compare two programmes, HIT and LIT, the magnitude in differences in avoided costs and QALY gains will remain

the same if we assume either no risk for relapse or the same risk for both programmes. The relapse assumptions are now presented in the Methods part (p.12, line 8-9) and Discussion (p.20, line 13-14).

3. It is not clear why the Markov model is until age 95 or death. A 5-8 year time frame for the cost-effectiveness would be a welcome addition to the literature.

The time horizon until age 95 or death was chosen according the recommendation from Sculpher at al: “Analysis should continue until the differences between options in terms of costs and effects (and hence cost effectiveness) are stable and unlikely to alter significantly”¹. The majority of Markov models used in evaluations in smoking cessation apply a life-time horizon² because current smoking influences future health risks and a smoking cessation today will reduce smoking-related health risks gradually during a long period. According to Bolin, 2012³, a lifetime perspective is necessary to estimate the total effects of smoking cessation. We add the reason in choosing the time horizon until age 95 or death to the manuscript (p. 12, line 6-8).

4. The distinction between the short-term CEA and long-term CEA is not well described. Is the only difference between the two models the quit rates? Are there assumptions in the short-term CEA about relapse rates?

We did not use relapse rates neither in the short-term CEA nor in the long-term CEA because the model does not contain the risks for relapse to smoking among the quitters. The relapse assumption and the limitations are now presented in the Methods part (p.12, line 8-9) and Discussion (p.20, line 13-14).

We used different outcomes in the short-term CEA and long-term CEA:

In short-term CEA: we used number and characteristics of 6-month continuous abstinent participants according to 1-year follow-up.

In long-term CEA: we used number and characteristics of sustained abstinent participants since planned smoking cessation date according to 5–8 years follow-up.

The distinction is now clarified on page 9, line 7-13 in the Methods section.

1. Sculpher M, Fenwick E, Claxton K. Assessing quality in decision analytic cost-effectiveness models. A suggested framework and example of application. *PharmacoEconomics* 2000;17(5):461-77.
2. Berg ML, Cheung KL, Hiligsmann M, et al. Model-based economic evaluations in smoking cessation and their transferability to new contexts: a systematic review. *Addiction* 2017;112(6):946-67. doi: 10.1111/add.13748
3. Bolin K. Economic evaluation of smoking-cessation therapies: a critical and systematic review of simulation models. *PharmacoEconomics* 2012;30(7):551-64. doi: 10.2165/11590120-000000000-00000

VERSION 2 – REVIEW

REVIEWER	Matthew Taylor Director, York Health Economics Consortium, University of York, UK
REVIEW RETURNED	25-Jun-2019
GENERAL COMMENTS	I thank the authors for addressing the majority of my queries regarding the previous draft. However, I still have two outstanding concerns:

	1. In the response to my ('Rev 1') second comment, the authors emphasise that "The main purpose of the study was to compare HIT and LIT programmes based on short- and long-term follow up and to check if cost-effectiveness based on short-term outcomes is sustainable in long-term". However, the conclusion section in the abstract focusses only on whether or not HIT is cost-effective and appears to be recommending an intervention. This needs to be clarified and made consistent with the objectives. 2. I think that Table 3 is still rather misleading. Where there are no participants in certain age groups, the row is left blank. This might suggest that there is 'no difference' between the groups, whereas in fact it just means that there is no evidence one way or the other. The blank cells should be replaced with 'not applicable'.
--	---

VERSION 2 – AUTHOR RESPONSE

Answers to the reviewer.

Rev. 1

1. In the response to my second comment, the authors emphasise that "The main purpose of the study was to compare HIT and LIT programmes based on short- and long-term follow up and to check if cost-effectiveness based on short-term outcomes is sustainable in long-term". However, the conclusion section in the abstract focusses only on whether or not HIT is cost-effective and appears to be recommending an intervention. This needs to be clarified and made consistent with the objectives.

Thank you for the comments. In the Discussion part, p 18, line 24-25, we stated:

“We also validated the cost-effectiveness results of the previous study based on short-term follow-up”, Further, we compared the results based on short-and long-term outcomes, p 18, line 28-30: “The incremental cost-effectiveness ratios (ICERs) were €918 and €3,786 using short- and long-term effectiveness, respectively, which are below the Swedish willingness-to-pay threshold of €50,000 per QALY “. Then on p. 19, line 1-2: “The results also confirm the conclusions of the previous cost-effectiveness analysis based on short-term follow-up data and suggest its sustainability” . We have also changed the Conclusions, p.21-22:

“In conclusion, the more costly HIT smoking cessation programme is an economically attractive option when compared to the LIT programme over a broad range of assumptions, using shot- and long-term outcomes”

2. I think that Table 3 is still rather misleading. Where there are no participants in certain age groups, the row is left blank. This might suggest that there is 'no difference' between the groups, whereas in fact it just means that there is no evidence one way or the other. The blank cells should be replaced with 'not applicable'.

Thank you, we replaced the blank cells with 'na – not applicable' in the Table 3.